# Refinement of the Magnesium–Aluminium Alloy Microstructure with Zirconium

**DOI:** 10.3390/ma15248982

**Published:** 2022-12-15

**Authors:** Cezary Rapiejko, Dominik Mikusek, Bartłomiej Januszewicz, Krzysztof J. Kubiak, Tadeusz Pacyniak

**Affiliations:** 1Faculty of Mechanical Engineering, Lodz University of Technology, Stefanowskiego 1/15, 90-537 Lodz, Poland; 2School of Mechanical Engineering, University of Leeds, Woodhouse Lance, Leeds LS2 9JT, UK

**Keywords:** magnesium alloy AZ91, zirconium, grain refinement, solidification process, investment casting

## Abstract

The magnesium–aluminium alloy AZ91 was inoculated with zirconium to refine the microstructure. Six different concentrations of zirconium content were tested, ranging from 0.1 to 0.6 wt %, and compared to the baseline AZ91 alloy without modification. Melted metal was poured into a preheated ceramic mould and the temperature was measured and recorded during the solidification. The derivative and thermal analysis (DTA) was performed to compare the crystallisation dynamics. Formed microstructure was analysed using an optical microscope, scanning electron microscopy (SEM-EDX) and energy dispersive X-ray spectrometry (XRD). The chemical composition was measured using an arc spectrometer. The time of solidification was shortened for the samples with a concentration of zirconium 0.3 wt % and the microstructure was refined. The level of grain refinement remained below 10% and the grain shape was changed to more spherical shapes. Both the primary magnesium and eutectic phases were modified. However, at a low concentration of zirconium (0.1 and 0.2 wt %), the primary grain size was increased. Therefore, the optimal zirconium concentration was 0.3 wt %. Larger concentrations (0.4 to 0.6 wt %) did not provide any additional benefit. Theoretical analysis showed that some Al_3_Zr intermetallic phases can form, which was confirmed on the derivate curve of the thermal analysis, and SEM-EDS and XRD analyses.

## 1. Introduction

Soring energy prices are stimulating the need for ever more efficient systems and use of lighter materials. Development of lightweight composite materials has seen significant progress in the last decade. However, due to poor recyclability and lack of long-term sustainability, their use is limited [1]. Lightweight aluminium-based metallic alloys are a very promising group of materials, but they are not suitable for high-strength applications [2]. Titanium-based alloys are too expensive and have poor tribological properties [3]. A promising group of materials are high-strength magnesium-based alloys that are affordable and sustainable in the long term. Strength-to-weight ratio remains one of the most important parameters that defines the use of such materials in the transport industry [4]. Several heat treatment methods have been developed to control the microstructure of magnesium alloys and to improve their mechanical properties [5,6,7]. They require additional processes, and this can increase the overall CO_2_ footprint during the material production. Some variants of the thermal methods developed recently, based on intensive cooling of the mould, can alleviate some of these problems [8,9]. However, use of inoculants is a much more efficient and less energy intensive way to improve the material properties at relatively low cost and in a sustainable manner [10,11,12,13]. If the amount of inoculant remains low, it may have negligible effect on a chemical composition, but it can significantly influence the grain size and the microstructure, leading to significant improvements in the mechanical and tribological properties [14]. Here, we should note the study of Yu Zhang et al. [11], who observed that a copper addition to the concentration within the range of 0.8% to 2.0% makes the eutectic phase α_Mg_ + γ(Mg_17_Al_12_) better ordered, and its arrangement becomes totally dendritic. Despite some interest in the inoculant effect on magnesium alloys [10,11,12,13,14,15,16,17,18,19] the mechanism of interactions and their effect are not fully understood.

The authors of the studies [20,21,22] confirmed that the effect of zirconium as an inoculant on the magnesium alloy microstructure containing aluminium is controversial and has not been fully explained. Magnesium alloys containing aluminium are usually used in industrial applications, and so examining the effect of modifying these alloys with zirconium is necessary. According to Hassan Jafari et al. [23], modifying the liquid alloy with zirconium (Zr) improves the properties of magnesium alloys due to the formation of the intermetallic phases Al_2_Zr. However, in their work, A. V. Koltygin et al. [24] showed that introducing zirconium into the alloy in a concentration of over 0.6% does not significantly change the mechanical properties of the modified alloy. A. V. Koltygin et al. [25], in another study, demonstrated that in magnesium alloys with iron and silicon impurities, segregation of zirconium takes place, which can have a detrimental effect on the mechanical properties of the final casts. M. Qian and A. Das [26] showed in their study that zirconium additions ranged from 0.2% to 0.3% lead equiaxed grains ahead of the columnar front. J.D. Robson and C. Paa-Rai [27] in their research were focused on grain refinement and ageing in magnesium–zinc–zirconium alloys. They concluded that those alloys cannot be hardened because Zn_2_Zr intermetallic phases could reduce strength by around 15%.

Based on the literature analysis, we can state that the effect of zirconium as an inoculant on the microstructure of magnesium alloys containing aluminium has not yet been sufficiently explained. The controversy around the interaction of zirconium with aluminium contained in magnesium alloys remains. That is why the authors have attempted to examine the influence of a small amount of zirconium introduced into alloy AZ91 on the crystallization process and the microstructure of the cast.

## 2. Materials and Methods

This research focuses on magnesium microstructure modification, and AZ91 alloy was selected as a base material. The chemical composition of the base magnesium alloy that was measured by the arc spectrometer SPECTROMAXx-Spectro is listed in Table 1. It conforms to the standard PN-EN 1753:2001 [28].

During the design of the experiment, seven magnesium alloy AZ91 melt samples were selected and prepared with different zirconium inoculant content, ranging from 0.1 to 0.6 wt % of the zirconium. All the melts have been prepared in an identical manner to avoid any unintended differences.

Each sample was melted in a crucible placed in a furnace SNOL 8.2/1100 OMEGA AB with resistance-based heating elements. The furnace was kept at a temperature of 740 ± 5 °C for 60 min. To prevent oxidation of melted magnesium alloy, sulphur powder was added. The zirconium inoculant was measured and added. The melt was kept in the furnace for another 30 min with the mixing spatula turned on to create a homogenous inoculant distribution. The cast was made in a ceramic mould that was preheated to 180 °C and fitted with DTA samplers. Inside the DTA samplers an S-type thermocouple (Pt-PtRh10) was placed to measure and record temperature variation during the solidification process. The melt schedule has been presented in Table 2.

The ceramic DTA samplers were made using investment casting technology, as described in the paper [29]. Investigation of the solidification as well as crystallisation processes of the modified alloys were carried out using the DTA method. The details of the procedure have been described in the articles [8,9].

The analysis of the crystallisation process t = f(τ), and its dynamics dt/dτ = f’(τ) have been performed using the DTA methodology. On the derivation curve described as dt/dτ = f’(τ), the following points were determined from the resulting thermal effects describing the formation of the corresponding phases in the microstructure of the alloys examined:

Points: Pk–A–D—represent formation of the primary phase, α_Mg_,

Points: D–E–F–H—represent formation of the eutectic phase α_Mg_ + γ(Mg_17_Al_12_).

At first, the chemical composition of the magnesium alloy samples were investigated using a SPECTROMAXx-Spectro (SPECTRO Analytical Instruments GmbH, Kleve, Germany) spark spectrometer. Furthermore, a more precise SEM-EDX analysis and X-ray diffraction (XRD) was performed. Identification of phases was carried out on Panalytical Empyrean diffractometer (Malvern Panalytical Netherlands, Almeo, The Netherlands) with symmetrical scan. The source of the X-rays was a tube with cobalt anode. It emited characteristic radiation CuKα = 1.54 Å working at 45 kV and 40 mA. The primary beam optic consisted of programmable divergence slit (PDF) ½ deg, fixed anti scatter slit 1°, Soller slit of 0.02 rad and a mask with size of 10 mm. The diffracted beam optics consisted of X’Celerator detector and Soller slits 0.02 rad. The diffraction pattern was obtained with step size 0.02 deg and counting time 200 s in the 2Theat range of 30–110 deg. Qualitative analysis was completed with the use of Panalytical High Score Plus Software ver. 3.0 and ICCD PDF4+ database. Identified phases are: Mg-Al solid solution—card no 04-017-4914; intermetallic compound Mg_17_Al_12_—card no 04-003-2934; and intermetallic compound Al_3_Zr—card no 01-074-5295.

To assess the microstructure of the magnesium alloys examined, the samples were etched in a solution that contained: 1 mL of acetic acid, 50 mL of distilled water and 150 mL of ethanol, and then were polished to a mirror finish. The microstructures of the obtained samples were observed under an optical microscope Nikon Eclipse MA 200, (Nikon, Tokyo, Japan). The microstructure was then subjected to image analysis and statistical analysis, which was carried out using “NIS Elements” (version 3.00, Tokyo, Japan) software.

Results obtained from the DTA analysis and the statistical analysis of the microstructure for the modified samples were compared to the normalised AZ91 magnesium alloy.

The DTA characteristics that were recorded and the values of time, temperature and kinetic values (derived values) for each of the particular characteristic points of new microstructure crystallisation have been established for the examined alloys. The DTA data provided the base for the solidification times for the primary phase αMg and the eutectic α_Mg_ + γ(Mg_17_Al_12_) calculations. The following equations were used to calculate the crystallisation times of the occurring phases:the phase α_Mg_: Δ*τ_α_* = *τ_D_* − *τ_pk_*,the eutectic α_Mg_ + γ(Mg_17_Al_12_): Δ*τ_γ_* = *τ_H_* − *τ_D_.*

To investigate the effects of zirconium modification on the crystallisation process of the magnesium alloy under study, based on DTA data with respect to the unmodified AZ91 alloy, the differences observed in the crystallisation time of the primary α_Mg_ and eutectic α_Mg_ + γ(Mg_17_Al_12_) phases were determined using the following equations:a difference in solidification times of the primary phase α_Mg_: Δτ_Dα_= Δτ_αAZ91_ − Δτ_αAZ91inX_,a difference in solidification times of the eutectic phase α_Mg_ + γ(Mg_17_Al_12_):
Δτ_Dγ_ = Δτ_γAZ91_ − Δτ_γAZ91inX_.
where: *_inX_* denotes the amount of Zr inoculant used.

To analyse the effect that the zirconium inoculant content can have on the mean sizes of the precipitation phase α_Mg_ and the eutectic phase α_Mg_ + γ(Mg_17_Al_12_), measurements of the mean perimeter as well as the diameter values of the given phases were conducted. The difference in the perimeters Δ*P* of the analysed primary phases αMg and the eutectic α_Mg_ + γ(Mg_17_Al_12_) for the modified alloys in reference to the baseline alloy AZ91 were calculated using the following relation, expressed in the percentage change:(1)$$\Delta P=\frac{\bar{P_{in}}-\bar{P_{B}}}{\bar{P_{B}}}\cdot 100\%$$
where:

ΔP—is a calculated difference in perimeters, %;

$\bar{P_{in}}$—is an average perimeter of modified phases α_Mg_ and α_Mg_ + γ(Mg_17_Al_12_), µm;

$\bar{P_{B}}$—is an average perimeter of phases α_Mg_ and α_Mg_ + γ(Mg_17_Al_12_) of the AZ91 alloy, µm.

In addition, change in a diameter Δ*D* for a given phase was calculated using the following formula:(2)$$\Delta D=\frac{\bar{D_{in}}-\bar{D}}{\bar{D_{B}}}\cdot 100\%$$
where:

ΔD—is a calculated difference in diameters, %;

$\bar{D_{in}}$—is an average diameter of modified phases α_Mg_ and α_Mg_ + γ(Mg_17_Al_12_), µm;

$\bar{D_{B}}$—is an average diameter of phases α_Mg_ and α_Mg_ + γ(Mg_17_Al_12_) of AZ91, µm.

## 3. Results and Discussion

### 3.1. Derivative and Thermal Analysis of Crystallisation Process Dynamics

The examples of DTA characteristics of the baseline magnesium alloy AZ91 and a modified alloy with 0.3 wt % of Zr are presented in Figure 1, while Table 3 contains the coordinates of characteristic points and their values for the AZ91 baseline alloy. Similar characteristics can be noted on both graphs with some additional peaks appearing at around 90 to 100 s in the case of the modified alloy; see Figure 1b. Small additional peaks can potentially be linked to the compound created between aluminium and zirconium, most likely to be Al_3_Zr. However, Al_3_Zr was not directly detected with the DTA method, and a hypothesis can be formed that this phase was trapped in the region close to the mould walls. This will be investigated further in Section 3.4. Table 4 contains the coordinates of the characteristic points on the DTA graphs (Figure 1) and their values for alloy AZ91 with 0.3 wt % of Zr.

The results of the DTA analysis, carried out according to the procedure described in the methodology section, are presented in Figure 2.

It can be observed in Figure 2 that the introduction of zirconium into AZ91 alloy changes the crystallisation time of both the primary phase α_Mg_ and the eutectic phase α_Mg_ + γ(Mg_17_Al_12_). The crystallisation time of the primary phase α_Mg_ has been shortened in all cases from 0.1 to 0.6 wt % of Zr. The most significant changes were observed for 0.1 and 0.3 wt % of Zr, and the times were −20.5 s and −16.0 s. This suggests that more nucleant particles have been formed around zirconium-rich regions, leading to faster crystallisation and therefore shorter crystallisation times. In the case of the eutectic phase α_Mg_ + γ(Mg_17_Al_12_) the crystallisation time is almost the same for 0.1 wt % of Zr and slightly longer of 5.1 s for 0.2 wt % of Zr. This indicates that there is no significant effect at this concentration of zirconium and, most likely, zirconium forms a stable compound with aluminium, iron, and other impurities. However, some effects of shorter crystallisation times can be observed for zirconium content between 0.3 and 0.6 wt % of Zr. This suggests that residual solute zirconium helped to create more nucleate particles, leading to potential grain refinement.

### 3.2. Microstructure Analysis

Metallographic studies were performed on the alloy samples under investigation. For all of the obtained microstructures, the images were taken, which were later subjected to an image analysis. Figure 3 shows examples of the microstructure of the AZ91 (Figure 3a,b) alloy and that of the modified AZ91 alloy with 0.3 wt % of Zr (Figure 3c,d) and 0.6 wt % of Zr (Figure 3e,f). It can be noted that the introduction of zirconium has a significant effect on both the size and the shape of the eutectic α_Mg_ + γ(Mg_17_Al_12_) precipitation. As can be observed in Figure 3d,f, the eutectic phase is forming more round-looking shapes. There is also a clear tendency for the primary phase α_Mg_ to form more spherical shapes (Figure 3c).

### 3.3. Image Analysis

Figure 4 shows representative examples of statistical analyses performed on the microstructure of AZ91 (Figure 4a,b), AZ91 alloy with 0.3% of Zr (Figure 4c,d) and AZ91 alloy with 0.6% of Zr (Figure 4e,f). The aim of these analyses was to compare the mean change in the grain size of primary phase α_Mg_ and the precipitation size of eutectic phases α_Mg_ + γ(Mg_17_Al_12_) in reference to the non-modified AZ91 alloy.

Figure 5 shows changes in the mean value of the grain perimeters ΔP observed after magnesium alloy modification with zirconium. The perimeter index was calculated according to Equation (1).

From Figure 5, we can observe that, for the inoculant concentrations of 0.1 and 0.2 wt % of Zr, the mean perimeters of eutectic phase precipitation α_Mg_ + γ(Mg_17_Al_12_) have increased, reaching the highest value of 8.8% for the zirconium concentration of 0.2 wt %. For the other examined concentrations, the precipitation of phase α_Mg_ + γ(Mg_17_Al_12_) were reduced, reaching the lowest value of −8.2% for the concentration of 0.3 wt % of Zr. In the case of phase α_Mg_, for all the examined zirconium concentrations, the mean precipitation perimeters were reduced, reaching a maximum reduction of −9.9% for the inoculant concentration equalling 0.3 wt % The smallest change in the mean perimeter value of −5.4% was observed for the zirconium concentration of 0.5 wt %.

Figure 6 shows the changes in the mean value of the grain diameter ΔD. The diameter index was calculated according to Equation (2).

As shown in Figure 6 the change in the mean diameter of the grains after the modification follows the same tendency as the one observed for the mean perimeter change. At zirconium concentrations of 0.1 and 0.2 wt %, the eutectic phase diameter is increasing and for the concentrations of 0.3 and above, the mean grain diameter is decreasing, leading to a more refined microstructure. In the case of primary phase α_Mg_, there is a clear tendency of grain refinement for all tested zirconium concentrations.

Overall, the grain refinement remains at relatively low level, below 10%. Therefore, the main question about the efficacy of the expensive zirconium as an inoculant for the Al-Mg alloy remains. The next part of the analysis will focus on the role of the aluminium when interacting with the zirconium inoculant.

To confirm the extent of the zirconium influence on the microstructure, SEM-EDX tests were performed. The presence of pure zirconium in the samples was very small, and only a trace amount was observed during the SEM and EDX analyses for the considered alloys. The zirconium concentration in the range up to 0.3 wt % was too low to influence the formation of any new phases or a significant change in any existing phase. Zirconium concentrations in the range of 0.3–0.6 wt % were observed in the zirconium-containing Al-Mn-(Mg) phases. This can be caused by the fact that zirconium dissolves in aluminium at a concentration of up to 0.28 wt %, whereas zirconium dissolves in magnesium at a concentration of up to 0.99 wt %. An exemplary image from the SEM-EDX analysis for AZ91 with 0.3 wt % of Zr is shown in Figure 7a. Considering the much better solubility of zirconium in magnesium, it can be assumed that zirconium dissolves in the primary phase α_Mg_. Figure 7b presents an image of the discussed phase, for which a chemical composition analysis was performed. An analysis diagram is shown in Figure 7c, whereas the analysis results are presented in Table 5.

To identify the observed zirconium phases in the microstructure of the AZ91 alloy, XRD analysis tests were carried out. The X-ray diffraction result of AZ91 with 0.3 wt % of Zr is shown in Figure 8.

The XRD results revealed that the main phases are: α-Mg matrix (Mg–Al solid solution), β-Mg_17_Al_12_ and Al_3_Zr. According to XRD analysis a small amount of the element Zr could exist in solid solution in the Mg matrix, and the excess Zr formed the Al_3_Zr phase, which is consistent with the Mg-Zr equilibrium phase diagram presented in next section.

### 3.4. Equilibrium Phase Diagram Analysis

The theoretical equilibrium phase diagram of Mg + 9 wt % of Al + 1 wt % of Zn + 0.0–0.5 wt % Zr was obtained using the Academic version of ThermoCalc software ver. 2022a. As can be seen in Figure 9, the first intermetallic compound formed at a very high temperature is Al_3_Zr_2_. It will later transform through several intermediate stages into a stable Al_3_Zr. Due to the high reactivity of aluminium and zirconium [30], such a compound can also be formed at lower temperatures of 740 °C as a first solid phase, and it will consume the zirconium inoculant. This can explain the low efficiency of the zirconium inoculant in Mg-Al alloys.

However, such a compound, if dispersed in the liquid melt, can potentially act as the nucleate particles, providing potential sites for a primary magnesium phase growth. This can explain the observed variation in the rate of temperature change on the derivate curve in Figure 1. However, it is also possible that other impurities present in the melt will form additional compounds with aluminium. Therefore, the level of purity of the Mg–Al alloys is of paramount importance to obtain a good quality, refined microstructure in the final product.

## 4. Summary

The magnesium–aluminium alloys remain a very popular lightweight material for casting. The AZ91 alloy is one of the most widely used. However, the high content of aluminium (9%) makes it more difficult to refine the grains and improve the mechanical properties. Zirconium is one of the very effective refinement inoculants for pure magnesium, but its effectiveness is very much restricted in Mg–Al alloys due to the fact that zirconium is very reactive with aluminium and can form stable Al_3_Zr intermetallic phases with a very high melting point. Therefore, they may fall to the bottom or remain close to the wall of the mould. Zirconium is also very susceptive to impurities and can easily form other compounds with Mn, Si, and Fe. Creation of the additional phases can be observed at early stages of primary phase crystallisation, as indicated in Figure 1b at times of 90–100 s. A small amount of the pure Zr could exist in solid solution in the Mg matrix, and the excess Zr formed Al_3_Zr phase. All those processes can lead to a change of Al content within the remaining melt before the crystallisation of the primary phase α_Mg_ and the eutectic phase α_Mg_ + γ(Mg_17_Al_12_) will begin.

From the results presented in this paper, we can notice a significant change in the shorter crystallisation times and in the smaller and more round-shaped grains that formed with an inoculant zirconium concentration of 0.3 wt % and above. Whereas the concentrations below 0.3 wt % (i.e., 0.1 and 0.2 wt % Zr) lead to larger grains of eutectic phase α_Mg_ + γ(Mg_17_Al_12_). Therefore, a zirconium inoculant concentration of 0.3 wt % is the most effective.

The shape of the grains follows the known tendency to form zirconium-rich cores that expand uniformly in all directions and create round or nodular grains of primary phase [31]. The eutectic phase solidifies in the remaining spaces between the primary phases, leaving both phases with smaller perimeters and diameters as indicated in Figure 5 and Figure 6.

Nevertheless, the efficiency of the zirconium inoculant remains quite low and can lead to small grain refinement below 10%. Further analysis of the Al_3_Zr intermetallic phase is required to better understand the mechanism of initial solidification and its influence on nucleant particles.

## 5. Conclusions

The AZ91 magnesium–aluminium alloy was modified by using a small amount of zirconium as an inoculant. A concentration range of 0.1–0.6 wt % of Zr was tested and the dynamics of the crystallisation process was analysed using Derivative and Thermal Analysis during solidification in a small mould. The grain sizes were analysed using an optical microscope and material composition was tested using an arc spectrometer, SEM-EDX and XRD techniques. Obtained results indicate small grain refinement below 10% for the inoculant concentration of 0.3 wt % Smaller concentration of zirconium leads to larger grain size compared to baseline material and a larger concentration above 0.3 wt % does not bring any additional benefits. From the results presented in this research, the following conclusions can be drawn:There is a change in the solidification mechanism at 0.3 wt % of Zn, with larger grains formed at lower concentrations and smaller grains formed at 0.3 wt % of Zn and above,Crystallisation time is shorter when grains are refined at the zirconium concentration above 0.3 wt %,Larger concentration of zirconium inoculant above 0.3 wt % are not beneficial,In early stages of primary magnesium phase solidification, additional peaks have been found on the temperature derivative curve (Figure 1), indicating the formation of additional intermetallic Al_3_Zr phases, which were directly detected in the bulk material.

## Figures and Tables

**Figure 1 materials-15-08982-f001:**
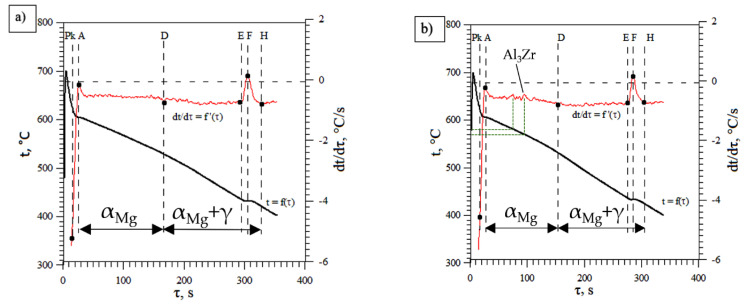
DTA characteristics of alloy AZ91 (**a**) and AZ91 with 0.3 wt % of Zr (**b**) during solidification in a ceramic DTA sampler.

**Figure 2 materials-15-08982-f002:**
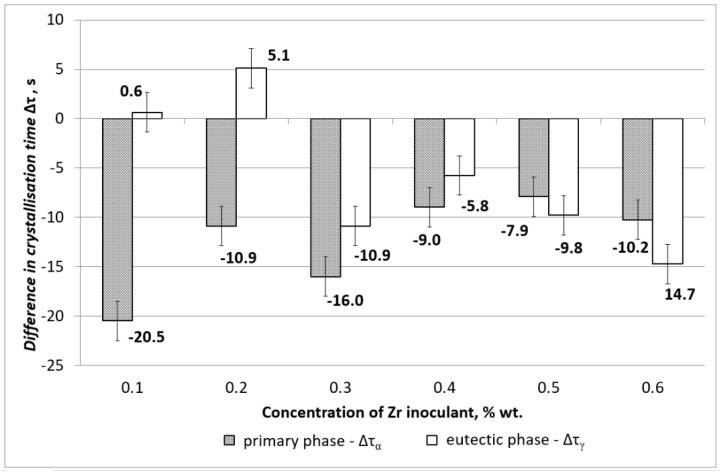
Difference in the solidification time of phase α_Mg_ and eutectic α_Mg_ + γ(Mg_17_Al_12_) of the Zr-inoculated magnesium alloys in reference to the baseline AZ91 alloy.

**Figure 3 materials-15-08982-f003:**
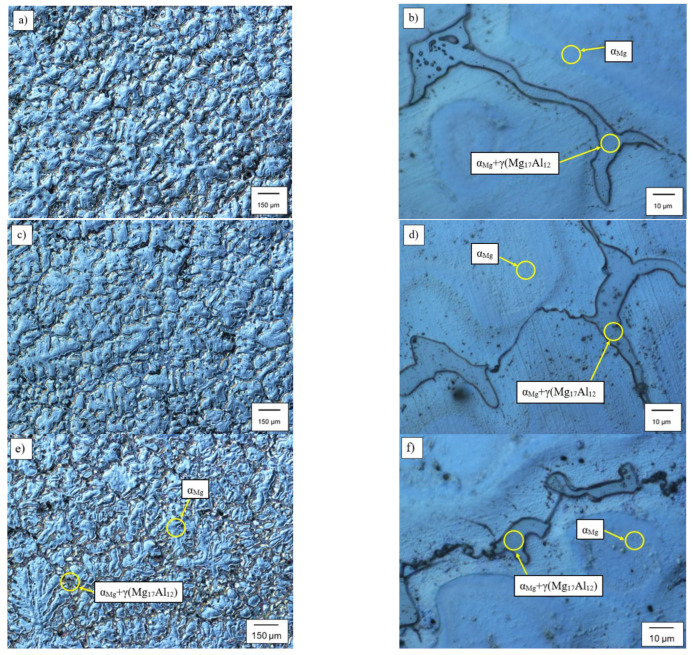
Microstructure of alloy: (**a**,**b**)—base magnesium alloy AZ91; (**c**,**d**)—modified magnesium alloy AZ91 with 0.3 wt % of Zr and (**e**,**f**)—modified magnesium alloy AZ91 with 0.6 wt % of Zr.

**Figure 4 materials-15-08982-f004:**
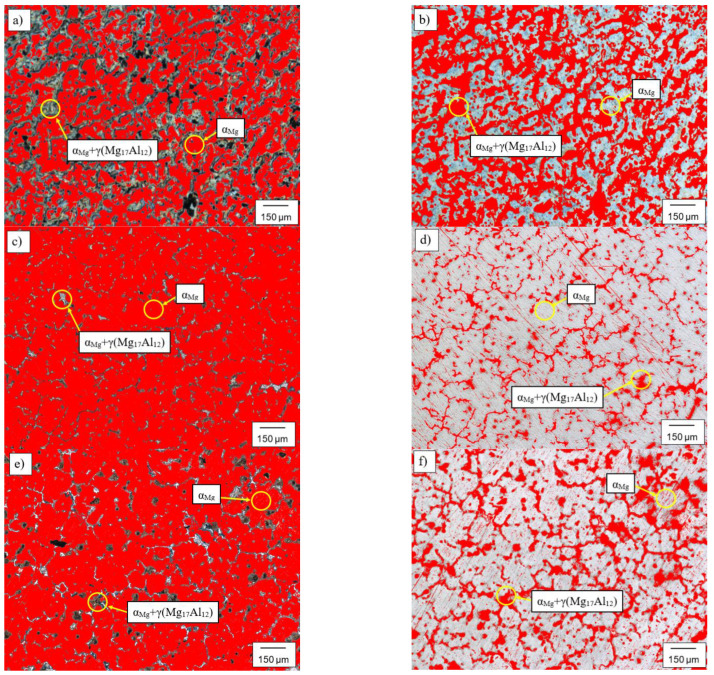
Microstructures subjected to a statistical image analysis of: (**a**,**b**)—alloy AZ91 subjected to a statistical image analysis:, (**c**,**d**)—microstructure of AZ91 alloy with 0.3 wt % Zr and (**e**,**f**)—microstructure of AZ91 alloy with 0.6 wt % Zr.

**Figure 5 materials-15-08982-f005:**
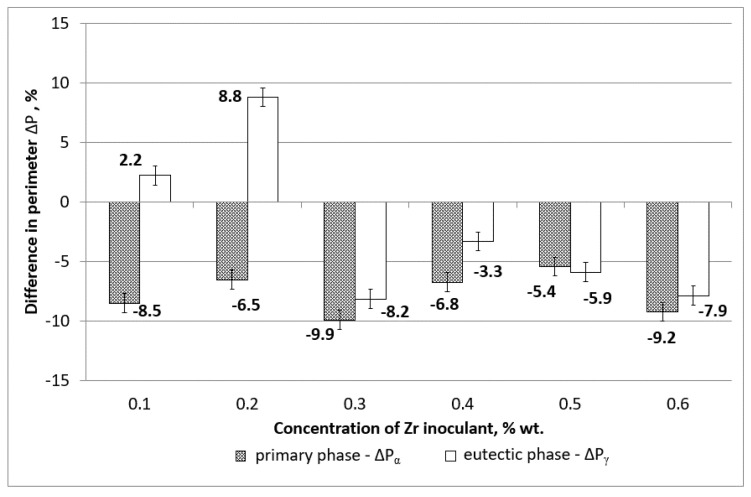
Change of the grain perimeters of phase α_Mg_ and α_Mg_ + γ(Mg_17_Al_12_) in samples inoculated in reference to AZ91.

**Figure 6 materials-15-08982-f006:**
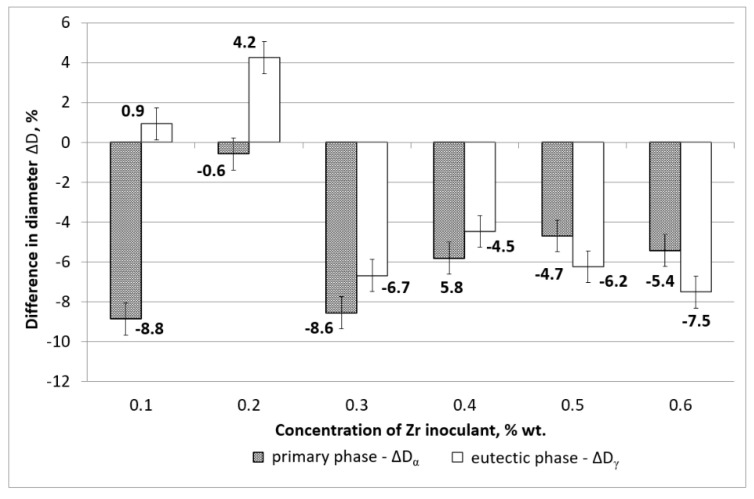
Change in the mean diameters of phase α_Mg_ and α_Mg_ + γ(Mg_17_Al_12_) in inoculated samples in reference to AZ91 baseline alloy.

**Figure 7 materials-15-08982-f007:**
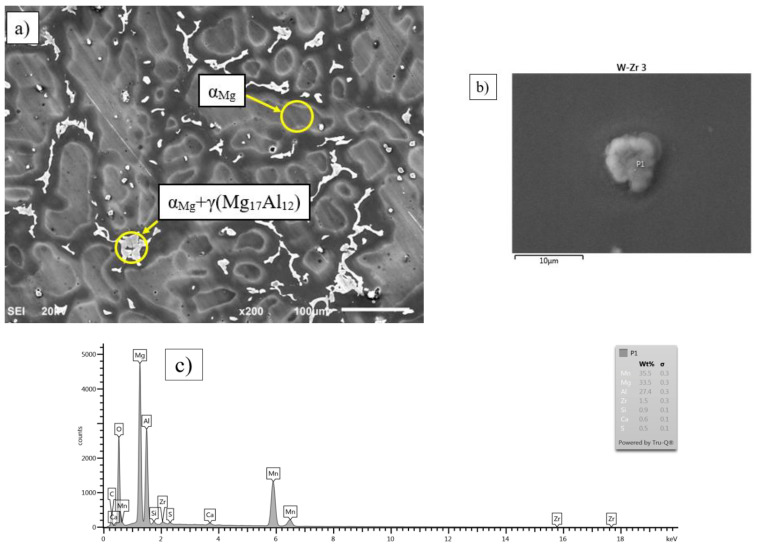
Microstructure of AZ91 with 0.3 wt % of Zr alloy during a SEM analysis (**a**), microstructure of α_Mg_ including a phase containing zirconium (**b**), and chemical composition spectrum of the analysed phase (**c**).

**Figure 8 materials-15-08982-f008:**
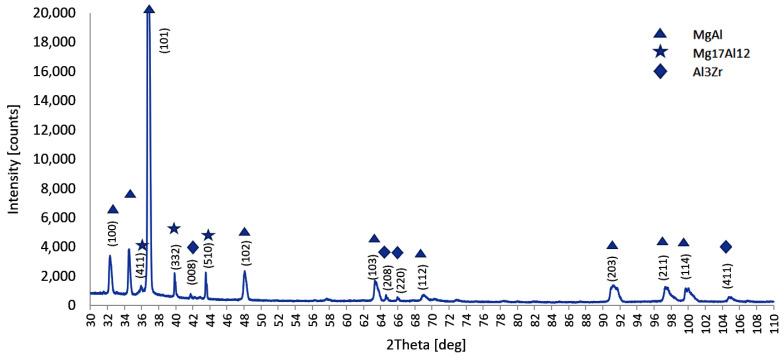
XRD spectra of AZ91 with 0.3 wt % of Zr.

**Figure 9 materials-15-08982-f009:**
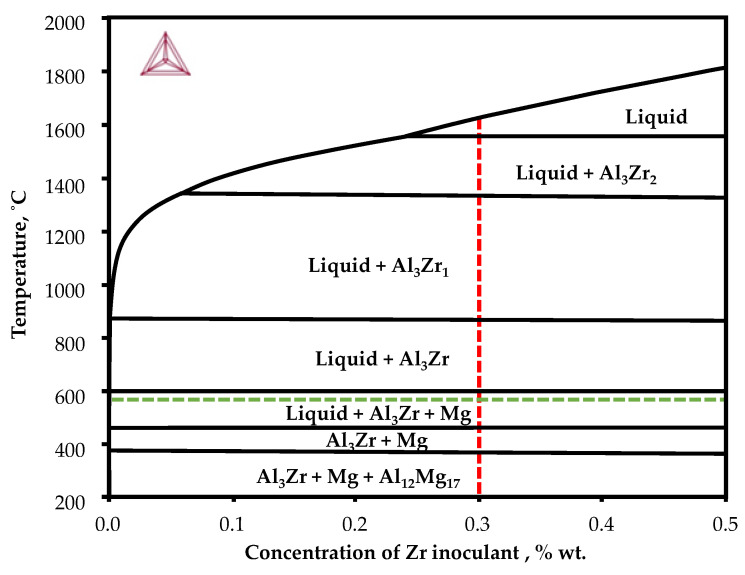
Theoretical phase diagram of Mg + 9 wt % of Al + 1 wt % of Zn + 0.0–0.5 wt % Zr calculated using ThermoCalc software.

**Table 1 materials-15-08982-t001:** The chemical composition of the baseline magnesium alloy AZ91.

Chemical Composition, wt %					
Mg	Al	Zn	Mn	Fe	Si
90.08	8.47	0.52	0.18	0.01	0.02

**Table 2 materials-15-08982-t002:** Prepared magnesium alloy melt samples with different zirconium inoculant content.

Sample	Zirconium Inoculant Content, wt %
Baseline	0.0
1	0.1
2	0.2
3	0.3
4	0.4
5	0.5
6	0.6

**Table 3 materials-15-08982-t003:** Characteristic DTA points of AZ91 magnesium alloy.

Point	τ, s	t, °C	dt/dτ, °C/s	Crystallising Phase
Pk	14.7	631.3	−5.2332	α_Mg_
A	26.2	604.4	−0.1646	
D	167.7	527.8	−0.7499	
E	293.1	436.6	−0.7371	α_Mg_ + γ(Mg_17_Al_12_)
F	305.3	432.1	0.1340	
H	329.0	420.2	−0.8102	

**Table 4 materials-15-08982-t004:** Characteristic DTA points of modified magnesium alloy AZ91 with 0.3 wt % of Zr.

Point	τ, s	t, °C	dt/dτ, °C/s	Crystallising Phase
Pk	17.3	622.8	−4.5792	α_Mg_
A	26.2	605.9	−0.2219	
D	154.2	530.9	−0.7911	
E	275.2	436.3	−0.7298	α_Mg_ + γ(Mg_17_Al_12_)
F	284.8	432.8	0.1443	
H	304.6	424.6	−0.7161	

**Table 5 materials-15-08982-t005:** Results of the chemical composition analysis of the AZ91 with 0.3 wt % of Zr taken at the point P1 in Figure 7b.

Element	wt %	wt % Sigma	Atomic, %
Mg	23.37	0.31	32.81
Al	30.06	0.36	38.03
Si	0.94	0.13	1.14
Ca	0.74	0.10	0.63
Mn	41.09	0.42	25.53
Fe	1.85	0.26	1.13
Zr	1.95	0.34	0.73
Total	100		100

## Data Availability

Not applicable.

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
