# Peer review of "Refinement of the Magnesium–Aluminium Alloy Microstructure with Zirconium"

_materials, 2022, doi:10.3390/ma15248982_

Round 1

Reviewer 1 Report

All comments in  attached file.

Author Response

Dear Reviwer,

Best regards

Cezary Rapiejko

Reviewer 2 Report

The abstract must undergo complete modification. Include the salient results.

on what basis the authors choose the Ziroconium content. What is the novelty in the work.

The sample dimension details are yet to mentioned. What is the test sample size used for DTA. The phase correlation must be complete

The microstrcture of All samples must be compared to analyse the results. The phase presence must be compared with XRD analysis. Do the authors did any phase fraction analysis?

Image J analysis is not an accurate method to find out the grain size. i suggest the authors to use EBSD datas

A structure property correlation must be there to interept the data to justify the Zr addition on to this alloy.

There are plenty of recent research available on the parent materials. Those paper can be reffered

Author Response

(The authors gave the same response as above.)

Reviewer 3 Report

1. The grammatical and typo errors must be revised.

2. It is recommended to discuss with more clarity on the spherical shaped microstructure with the addition of Zr.

3. To confirm the phases formed in the Mg-Al alloy, authors used only optical and SEM microstructure, but these investigations are not enough to confirm them. Therefore, I am recommending to perform XRD to confirm the phases.

4. Authors depicted all the characterizations only for 0.3 wt.% Zr Mg-Al alloys, why?  If authors wanted to show only 0.3 wt.% Zr Mg-Al alloys, then why did they used 0.1, 0.2, 0.4, 0.5 and 0.6 wt % Zr additions? It seems baseless.

Author Response

(The authors gave the same response as above.)

Round 2

Reviewer 2 Report

Dear Authors, 

I appreciate your effort in improving the manuscripts. However, I have noticed the XRD peak can be indexed with hkl values. Kindly consider this suggestion.

Author Response

Reviewer 2

I appreciate your effort in improving the manuscripts. However, I have noticed the XRD peak can be indexed with hkl values. Kindly consider this suggestion.

 Authors’ Response:  We thank the Reviewer for suggestion. According to suggestion we have added  the  hkl values on X Ray Diffraction Analysis.

Reviewer 3 Report

Dear Authors, Thank you for revising the manuscript as per my suggestions. Now the article is in great shape.

Author Response

Reviewer 3

Dear Authors, Thank you for revising the manuscript as per my suggestions. Now the article is in great shape.

 Authors’ Response:  We would like to thank the Reviewer for the positive comments.